REGISTERED REPORT

# Registered report: Interactions between cancer stem cells and their niche govern metastatic colonization

**Francesca Incardona[1], M Mehdi Doroudchi[1], Nawfal Ismail[1], Alberto Carreno[1], Erin Griner[2], Minyoung Anna Lim[3], Reproducibility Project: Cancer Biology***

[1]BTS Research, San Diego, California; [2]University of Virginia, Charlottesville, Virginia; [3]EMD Millipore, Temecula, California

**Abstract** The Reproducibility Project: Cancer Biology seeks to address growing concerns about reproducibility in scientific research by replicating selected results from a substantial number of high-profile papers in the field of cancer biology published between 2010 and 2012. This Registered report describes the proposed replication plan of key experiments from 'Interactions between cancer stem cells and their niche govern metastatic colonization' by Malanchi and colleagues, published in *Nature* in 2012 (*Malanchi et al., 2012*). The key experiments that will be replicated are those reported in Figures 2H, 3A, 3B, and S13. In these experiments, Malanchi and colleagues analyze messenger RNA levels of periostin (POSTN) in pulmonary fibroblasts, endothelial cells, and immune cells isolated from mice with micrometastases to determine which cell type is producing POSTN in the metastatic niche (Figure 2H; *Malanchi et al., 2012*). Additionally, they examine MMTV-PyMT control or POSTN null mice to test the effect of POSTN on primary tumor growth and metastasis (Figures 3A, 3B, and S13; *Malanchi et al., 2012*). The Reproducibility Project: Cancer Biology is a collaboration between the Center for Open Science and Science Exchange, and the results of the replications will be published in *eLife*.

***For correspondence:**
tim@cos.io

**Group author details**
Reproducibility Project: Cancer Biology
See page 13

**Competing interests:**
See page 13

## Introduction

Metastatic colonization is a highly inefficient process that only a small subset of disseminated tumor cells accomplish (*Nguyen et al., 2009*). A growing body of literature suggests that cancer stem cells (CSC), tumor cells with the ability to self-renew and differentiate, play important roles not only in metastatic colonization but also in establishing the metastatic niche to support metastatic colonization (*Visvader and Lindeman, 2012*). Using the MMTV-PyMT mouse breast cancer model, which spontaneously metastasizes to the lungs, Malanchi and colleagues reported that only the CSC population, identified as $CD24^+ CD90^+$, were capable of initiating lung metastases and secondary metastases (*Guy et al., 1992*; *Lin et al., 2003*; *Malanchi et al., 2012*). Additionally, only a subset of the injected CSC population resulted in metastatic nodules. Periostin (POSTN) was identified by microarray RNA profiling studies as a stromal factor involved in maintaining the normal stem cell niche and demonstrated to be secreted by stromal fibroblasts, but not by infiltrating tumor cells (*Malanchi et al., 2012*). POSTN is a secreted protein that is incorporated in the extracellular matrix and has been associated with metastasis in several human cancers (*Conway et al., 2014*). The functional necessity of POSTN was investigated by observing the pulmonary metastatic potential in POSTN-knockout MMTV-PyMT mice, which showed a statistically significant decrease compared to controls (*Malanchi et al., 2012*). This was further demonstrated by observing a rescue in metastatic efficiency by injecting POSTN-deficient tumor cells into wild-type recipient mice (*Malanchi et al., 2012*). Investigating the mechanism of POSTN-induced metastasis, Malanchi and colleagues reported a decrease in colony

formation in vitro with POSTN-deficient tumor cells or wild-type CSCs co-cultured with POSTN-deficient stromal cells, demonstrating the involvement of POSTN in stem cell maintenance (*Malanchi et al., 2012*). Furthermore, POSTN was reported to bind to Wnt ligands, leading to an increase in Wnt signaling in CSCs, a known regulator of stem cell maintenance in a variety of tissues (*Malanchi et al., 2012*). Taken together, these results suggest that CSCs are essential for metastatic colonization and that CSCs induce stromal fibroblasts to secrete POSTN in the metastatic niche to support tumor cell outgrowth by augmenting the Wnt signaling pathway.

Malanchi and colleagues' findings suggest that targeting of POSTN in the metastatic niche could potentially be used to treat metastasis. The key experiments included for replication were selected because they examine the induction of *POSTN* expression in the pulmonary stromal fibroblasts and test the role of POSTN in primary tumor formation and metastatic efficiency, which are relevant as the role of POSTN as a possible prognostic marker and target for anticancer therapies is explored (*Xu et al., 2012*; *Nuzzo et al., 2014*). Indeed, treatment of mice with POSTN specific DNA aptamers, single-stranded DNA oligonucleotides designed to bind and inhibit POSTN, was shown to decrease primary tumor growth and metastasis in a xenograft model of mammary tumorigenesis (*Lee et al., 2013*). Additionally, treatment of ovarian xenograft models with a neutralizing antibody to POSTN resulted in a reduction of metastatic potential and tumor growth, migration, and invasion (*Zhu et al., 2011*).

Figures 3A, 3B, and Supplemental Figure S13 examine the role of POSTN in primary tumor formation and metastasis utilizing a MMTV-PyMT mouse model of mammary tumor formation. This mouse model develops primary mammary tumors that spontaneously metastasize to the lung (*Guy et al., 1992*; *Lin et al., 2003*). $MMTV\text{-}PyMT^{+/tg}$; $Postn^{-/-}$ mice and their $Postn^{+/+}$ counterparts were analyzed for primary tumor formation and metastasis. Primary tumor size was measured by weight, and the number of metastases was determined by counting lung metastatic lesions. Malanchi and colleagues reported in Figures 3A and S13A that POSTN does not affect the size of the primary tumors, but $MMTV\text{-}PyMT^{+/tg}$; $Postn^{-/-}$ mice develop significantly fewer metastases than their $Postn^{+/+}$ counterparts (*Malanchi et al., 2012*). The importance of stromal POSTN was also demonstrated in another study in which orthotopic inoculation of gastric cancer cells into $Rag2^{-/-}$; $Postn^{-/-}$ mice reduced tumor size, decreased invasiveness, and decreased growth compared to $Rag2^{-/-}$; $Postn^{+/+}$ mice (*Kikuchi et al., 2014*). Additionally, overexpression of POSTN in human mammary epithelial and breast cancer cells resulted in enhanced tumor growth and metastasis (*Wang et al., 2013*), which is similar to a colon cancer cell model where overexpression of POSTN resulted in an increase in the number and size of liver metastases (*Bao et al., 2004*). The experiment reported in Figures 3A, 3B, and Supplemental Figure S13 will be replicated in Protocol 1.

Malanchi and colleagues show that POSTN is expressed primarily in fibroblasts and to a lesser extent in endothelial cells but is not expressed in immune cells at sites of metastasis (*Malanchi et al., 2012*). This was determined by FACS sorting cells from lungs with macrometastases to isolate CD34[+]/CD31[−] pulmonary fibroblasts, CD31[+] endothelial cells, and CD45[+] immune cells. Quantitative PCR of messenger RNA extracted from each of these cell populations was then used to determine the relative expression of *POSTN* in each cell type as reported in Figure 2H (*Malanchi et al., 2012*). RNA in situ hybridization and immunostaining showed similar results (*Malanchi et al., 2012*). Similarly, another study used immunohistochemistry and immunofluorescence to show that POSTN was localized to stromal fibroblasts in human samples of advanced gastric cancer (*Kikuchi et al., 2014*). Yet, another study utilized tandem mass spectrometry and immunofluorescence to find POSTN concentrated in the extracellular matrix surrounding sites of neovascularization of micrometastases and within the tips of endothelial cells involved in the neovascularization (*Ghajar et al., 2013*). The experiment reported in Figure 2H will be replicated in Protocol 2.

## Materials and methods

### Protocol 1: tumor size and metastases of $MMTV\text{-}PyMT^{+/tg}$; $Postn^{+/+}$ and $MMTV\text{-}PyMT^{+/tg}$; $Postn^{-/-}$ mice

This experiment examines the requirement of POSTN in metastatic colonization using the MMTV-PyMT mouse model. Female mice carrying the MMTV-PyMT transgene that are either $Postn^{+/+}$ or $Postn^{-/-}$ will be examined for changes in primary tumor size and the number of spontaneously formed pulmonary macrometastases, which is a replication of the experiment reported in Figures 3A, 3B, and

Supplemental Figure 13. This experiment will also generate lung tissue from *MMTV-PyMT*[+/tg]; *Postn*[+/+] female mice that are positive with metastatic disease for use in Protocol 2.

## Sampling

- Experiment has 2 cohorts:
  - ○ Cohort 1: *MMTV-PyMT*[+/tg]; *Postn*[+/+] female mice.
  - ○ Cohort 2: *MMTV-PyMT*[+/tg]; *Postn*[−/−] female mice.
- Experiment will use the following number of mice per cohort:
  - ○ Cohort 1: 14 mice.
  - ○ Cohort 2: 15 mice.
    - Note: Derive mice from consecutive litters and analyze development of tumors and metastases until the two cohorts reach the indicated numbers.
- To account for outlier data, as presented in the original publication, 5% and 10% more mice were added to each cohort, respectively, to ensure at least 13 mice survive each cohort for a minimum power of 80%.
  - ○ See 'Power calculations' section for details.

## Materials and reagents

| Reagent | Type | Manufacturer | Catalog # | Comments |
|---|---|---|---|---|
| *MMTV-PyMT*[+/tg]; *Postn*[+/−] FVB male mouse | Animal model | Original lab | n/a | From original lab |
| *MMTV-PyMT*[+/tg]; *Postn*[−/−] FVB male mouse | Animal model | Original lab | n/a | From original lab |
| *Postn*[−/−] FVB female mice | Animal model | Original lab | n/a | From original lab |
| *Postn*[+/+] FVB female mice | Animal model | Charles River | Strain code: 207 | Original was from France |
| PureGenome tissue DNA extraction kit | Kit | EMD Millipore | 72635 | Original not specified |
| MMTV-PYVT384 primer | Nucleic acid | Sequences provided by original authors; specific brand information will be left up to the discretion of the replicating lab and recorded later | | |
| MMTV-PYVT385 primer | Nucleic acid | | | |
| Postn-5′ primer | Nucleic acid | | | |
| Postn-3′ primer | Nucleic acid | | | |
| INT-as primer | Nucleic acid | | | |
| dNTPs (10 mM) | Chemical | Sigma–Aldrich | D7295 | Included during communication with authors. Original brand not specified |
| Taq-polymerase (with tubes of 10× PCR buffer and 25 mM MgCl$_2$) | Enzyme | Sigma–Aldrich | D4545 | Included during communication with authors. Original brand not specified |
| PCR system | Equipment | Applied Biosystems StepOne | | Original not specified |
| Isoflurane | Chemical | Specific brand information will be left up to the discretion of the replicating lab and recorded later | | |
| StereoZoom stereomicroscope, zoom range 0.8×–4.0× | Instrument | Bausch & Lomb | n/a | Original a Leica M205 FA |

## Procedure

1. Breed *MMTV-PyMT*[+/tg]; *Postn*[+/−] or *MMTV-PyMT*[+/tg]; *Postn*[−/−] male mice with *Postn*[+/+] and *Postn*[−/−] female mice to obtain *MMTV-PyMT*[+/tg]; *Postn*[+/+] control and *MMTV-PyMT*[+/tg]; *Postn*[−/−] experimental female mice, respectively.
   a. Do not use *MMTV-PyMT* female mice for breeding as they develop mammary tumors. Additionally, *MMTV-PyMT*[+/tg] males cannot be crossed with *MMTV-PyMT*[+/tg] females as the progeny will have a double dose of the oncogene and develop extremely aggressive tumors that cannot be used in this study.
   b. For generation of the *MMTV-PyMT*[+/tg]; *Postn*[−/−] experimental female mice, male *MMTV-PyMT*[+/tg]; *Postn*[−/−] male mice will be crossed with *Postn*[−/−] female mice.
   c. For generation of the *MMTV-PyMT*[+/tg]; *Postn*[+/+] control female mice, the *MMTV-PyMT*[+/tg]; *Postn*[+/−] male mouse will be crossed to *Postn*[+/+] female mice to obtain *MMTV-PyMT*[+/tg]; *Postn*[+/+] male mice that will then be crossed with *Postn*[+/+] female mice.

d. As mice obtained were found to contain agents, cross-foster rederivation of mice will be performed based on the following procedure (*Artwohl et al., 2008*). Donor dam will be placed with male mice 5 days a week Monday afternoon to Friday morning and plugs will be checked daily. When dam is found plugged, she should be single housed and date of plug will be recorded as a sign of pregnancy. Once the pregnancy is confirmed, a timed pregnant mouse will be ordered to use as a foster recipient. Foster recipients will ideally have a different fur coat color than the donor dam, so the identification of the fostered pups will be easier. Consequently, the donor and recipient dams will be removed from their cages and placed in separate clean cages. The litter to be fostered will gently be picked up and cleaned with alcohol and passed to a clean tech to be mixed with dirty bedding, nestlet, and other pups from the recipient dam's cage. When mixing the pups, they will be gently arranged in the palm of the hand, in contact with nestlet and bedding from the recipient dam's cage to transfer the recipient dam's scent. All pups will be placed back in the nest and the recipient dam will be returned to the cage. The cage will be monitored visually every 15 min for the first hour; if there is evidence of rejection by the dam (agitation, carrying the pups around), the pups will be removed from the cage and humanely euthanized. The cages will be visually assessed at least twice daily and will not be disturbed for the first 72 hr after fostering in order to avoid any potential cannibalism. All experimental animals will not be treated with Ivermectin or Fenbendazole, as these could change a number of immune parameters affecting tumor growth and take rate.

2. Extract genomic DNA from mouse tail snips using DNA extraction kit following manufacturer instructions.
    a. From manufacturer's instructions follow 'Solid Tissue' assay protocol.
3. Genotype mice by PCR with *MMTV-PyMT* and *Postn* primers.
    a. MMTV-PYVT384 primer: GGA AGC AAG TAC TTC ACA ACG G.
        i. This primer is one nucleotide different than what is listed on the Jackson Laboratory information for stock number 002374 (FVB/N-Tg(MMTV-PyVT)634Mul/J), but is used successfully by the original lab.
    b. MMTV-PYVT385 primer: GGA AAG TCA CTA GGA GCA GGG.
    c. Postn-5′ primer: GGT GCT TCT GTA AGG CCA TC.
    d. Postn-3′ primer: GTG AGC CAG GAC CTT GTC ATA.
    e. INT-as primer: AGC ACT GAC TGC GTT AGC AA.
    f. Genotyping will be determined by examining both amplicon size and presence.
        i. MMTV-PyMT conditions (oncogene = 556 bp band):

| | |
|---|---|
| 10× PCR buffer | 1.50 µl |
| 50 mM MgCl$_2$ | 0.45 µl |
| 10 mM dNTPs | 0.30 µl |
| MMTV-PYVT384 primer | 0.10 µl |
| MMTV-PYVT385 primer | 0.10 µl |
| Taq-polymerase | 0.20 µl |
| H$_2$O | Bring up to 13 µl |
| DNA (1:20 dilution) | 2 µl |

Cycling parameters:
1. 94°C pause.
2. 94°C for 3 min.
3. 12 cycles of:
    i. 96°C for 20 s.
    ii. 64°C for 30 s.
    iii. 72°C for 65 s.
4. 25 cycles of:
    i. 94°C for 20 s.
    ii. 58°C for 30 s.
    iii. 72°C for 35 s.
5. 72°C for 2 min.
6. 20°C pause.

ii. *Postn* conditions (WT = 245 bp band; KO = 182 bp band):

| | |
|---|---|
| 10× PCR buffer | 1.50 µl |
| 50 mM MgCl$_2$ | 0.45 µl |
| 10 mM dNTPs | 0.30 µl |
| Postn-5′ primer | 0.10 µl |
| Postn-3′ primer | 0.10 µl |
| INT-as primer | 0.10 µl |
| Taq-polymerase | 0.20 µl |
| H$_2$O | Bring up to 13 µl |
| DNA (1:20 dilution) | 2 µl |

Cycling parameters:
1. 94°C pause.
2. 94°C for 1 min.
3. 45 cycles of:
    i. 96°C for 6 s.
    ii. 59°C for 20 s.
    iii. 72°C for 30 s.
4. 20°C pause.

4. Separate and image amplicons by agarose gel electrophoresis.
5. Monitor *MMTV-PyMT*$^{+/tg}$; *Postn*$^{+/+}$ and *MMTV-PyMT*$^{+/tg}$; *Postn*$^{-/-}$ female mice for tumor development and keep until tumor disease is fully developed and the metastatic disease is estimated to occur.
    a. Record age of mice when palpable tumors are detected.
        i. Multiple tumors will form and grow until they reach a significant size.
    b. Monitor health status of mice. If mice have to be euthanized prior to reaching fully developed primary tumors exclude mice from study and record reason for euthanasia.
    c. Tumor disease is fully developed when primary tumors have developed in all mammary glands with an average weight around 1 g per tumor.
        i. Both cohorts of mice will develop tumors approximately within 3–4 months of age and up to 6 months.
        ii. Record mice with tumors that are large and form close to the neck as these may give metastasis more efficiently.
    d. Record age of mice when sacrificed and determine time gap between detection and fully developed tumor.
        i. Method of euthanasia is isoflurane overexposure (2–5% at 1 l/min) followed by cervical dislocation.
6. Dissect primary tumor and lung tissue from mice.
    a. Weigh primary tumors.
        i. Record total weight of all primary tumors together for each mouse.
        ii. Record the number of primary tumors for each mouse.
        iii. Divide total weight of all primary tumors by number of primary tumors to obtain a reference primary tumor weight for each mouse.
7. Dissect lungs and blindly count the number of macrometastatic nodules on every side of all separated lobes of the lung using a stereomicroscope.
    a. Do not fix or stain tissues.
    b. Quickly count large macrometastatic nodules (≥1 mm in diameter) on all sides.
8. Immediately after counting macrometastasis, use the first six lungs identified from *MMTV-PyMT*$^{+/tg}$;*Postn*$^{+/+}$ female mice that are positive with metastatic disease for further analysis (Protocol 2).
    a. Should be approximately 5–6 months of age.
        i. Exclude lungs from mice that are euthanized before this age.
    b. Use mice with detectable metastatic disease and record number of macrometastases in each lung used and the total weight of the primary mammary tumors.

## Deliverables

- Data to be collected:
  - Mouse health records (age of palatable tumor detection, reason for early euthanasia and exclusion of mice, age of mice with fully developed tumor when sacrificed, mice with large tumors formed close to the neck).
  - Gel images of PCR genotyping (Compare to Figure S10).
  - Number of primary tumors formed and total weight of all tumors for each mouse.
  - Raw numbers and box and whisker plot of weight of primary tumors (total weight divided by number of primary tumors) for each mouse. (Compare to Figure 3A and S13).
  - Raw numbers of pulmonary macrometastases for each mouse (Compare to Figure 3B and S13).
  - Box and whisker plot of number of pulmonary macrometastases for each mouse. (Compare to Figure 3B and S13).
- Sample delivered for further analysis:
  - Lungs for FACS and qRT-PCR analysis for Protocol 2.

## Confirmatory analysis plan

The original paper reported outliers in each cohort of mice. As these appear to have not been included in the original analysis, we will also remove any outliers from the analysis for comparison. But the analysis will also be performed with all data values. Outliers are determined as 1.5 times the interquartile range.
   This replication attempt will perform the following statistical analysis listed below:

- Statistical analysis:
  - Primary tumor weight in $MMTV\text{-}PyMT^{+/tg}$;$Postn^{+/+}$ mice relative to $MMTV\text{-}PyMT^{+/tg}$;$Postn^{-/-}$ mice.
    - Unpaired two-tailed $t$-test.
  - Number of pulmonary macrometastases in $MMTV\text{-}PyMT^{+/tg}$;$Postn^{+/+}$ mice relative to $MMTV\text{-}PyMT^{+/tg}$;$Postn^{-/-}$ mice.
    - Unpaired two-tailed $t$-test.
  - The replication attempt will also perform negative binomial regression analysis of the macrometastases count data.
- Meta-analysis of effect sizes:
  - Compute the effect sizes of each comparison, compare them against the reported effect size in the original paper, and use a meta-analytic approach to combine the original and replication effects, which will be presented as a forest plot.

## Known differences from the original study

The mice in the original study were from an 8[th] generation backcross to the FVB line, thus it was on a mixed background, while the mice used in the replication will be from a pure 10th generation backcross to the FVB line. This may make a difference in the effect size and will be included in the discussion of the results of the replication. Additionally, mice will undergo cross-foster rederivation to attempt to remove agents currently associated with the mice. All known differences of materials and reagents are listed in the materials and reagents section above with the originally used item listed in the comments section. All differences have the same capabilities as the original and are not expected to alter the experimental design.

## Provisions for quality control

Mice will undergo cross-foster rederivation to attempt to remove agents currently associated with the mice that could alter immune parameters affecting tumor growth and take rate. All data obtained from the experiment—raw data, data analysis, control data, and quality control data—will be made publicly available, either in the published manuscript or as an open access data set available on the Open Science Framework project page for this study (https://osf.io/vseix).

## Protocol 2: POSTN expression in lung stroma with macrometastases

This experiment uses quantitative PCR to detect the expression level of $POSTN$ in CD34+/CD31− pulmonary fibroblasts, CD31+ endothelial cells, and CD45+ immune cells isolated from lungs of mice with macrometastases, which is a replication of the experiment reported in Figure 2H.

### Sampling

- Experiment will use six lungs for a minimum power of 82%.
  - See appendix for detailed power calculations.

- Each lung will be isolated into 3 cohorts:
  - Cohort 1: CD34$^+$/CD31$^-$ pulmonary fibroblasts from *MMTV-PyMT$^{+/tg}$;Postn$^{+/+}$* mice.
  - Cohort 2: CD31$^+$ endothelial cells from *MMTV-PyMT$^{+/tg}$;Postn$^{+/+}$* mice.
  - Cohort 3: CD45$^+$ immune cells from *MMTV-PyMT$^{+/tg}$;Postn$^{+/+}$* mice.
    - Each cohort will be sorted using the following antibodies:
      - CD45.
      - CD31.
      - CD34.
      - Isotype controls.
      - Unstained control.
- Each cohort will be analyzed for the following gene expression levels:
  - POSTN.
  - GAPDH.

## Materials and reagents

| Reagent | Type | Manufacturer | Catalog # | Comments |
|---|---|---|---|---|
| 50 ml tubes | Labware | Sigma–Aldrich | CLS430290 | Originally not specified |
| Hank's balanced salt solution (HBSS) | Buffer | Sigma–Aldrich | H6648 | Included during communication with authors. Original brand not specified |
| Liberase TM | Enzyme | Roche | 05401127001 | – |
| Liberase TH | Enzyme | Roche | 05401151001 | – |
| DNase | Enzyme | Sigma–Aldrich | DN25 | – |
| Phosphate buffered saline (PBS) without MgCl$_2$ and CaCl$_2$ | Buffer | Sigma–Aldrich | D8537 | Original brand not specified |
| EDTA | Chemical | Included during communication with authors. Specific brand information will be left up to the discretion of the replicating lab and recorded later | | |
| Bovine serum albumin (BSA) | Chemical | Sigma–Aldrich | A3803 | Included during communication with authors. Original brand not specified |
| 100 µm cell strainer | Labware | Corning | 431752 | Original brand not specified |
| 2.5 ml syringe | Labware | Included during communication with authors. Specific brand information will be left up to the discretion of the replicating lab and recorded later | | |
| Fetal bovine serum (FBS) | Cell culture | Sigma–Aldrich | F0392 | Original brand not specified |
| Polypropylene (opaque) FACS tubes | Labware | Specific brand information will be left up to the discretion of the replicating lab and recorded later | | |
| CD45 (clone 30-F11) PE-Cy5.5 antibody (rat IgG2b, kappa) | Antibodies | eBioscience | 35-0451-80 | Use at 1:300 |
| CD31 (clone 390) Pac.Blue antibody (rat IgG2a, kappa) | Antibodies | Invitrogen | RM5228 | Use at 1:200 |
| CD34 (clone RAM34) PE antibody (rat IgG2a, kappa) | Antibodies | BD Pharmingen | 551387 | Use at 1:50 |
| Rat IgG2b, kappa isotype control PE-Cy5.5 | Antibodies | eBioscience | 35-4031 | Use at 1:300 dilution. Originally not specified |
| Rat IgG2a, kappa isotype control Pac.Blue | Antibodies | Invitrogen | R2a28 | Use at 1:200 dilution. Originally not specified |
| Rat IgG2a, kappa isotype control PE | Antibodies | BD Pharmingen | 553930 | Use at 1:50 dilution. Originally not specified |
| 7AAD | Chemical | BioLegend | 420403 | Use at 1:1000. Original brand not specified |
| Flow cytometric cell sorter | Instrument | BD Pharmingen | FACSAria II | Original from Beckman Coulter |
| FlowJo | Software | – | – | – |

*Continued on next page*

*Continued*

| Reagent | Type | Manufacturer | Catalog # | Comments |
|---------|------|--------------|-----------|----------|
| TRI reagent | Chemical | Sigma–Aldrich | T9424 | Replaces RNA extraction kit from Qiagen |
| Oligo dT (18) | Nucleic acid | Life Technologies | SO132 | Included during communication with authors. Original brand not specified |
| Oligo dT (23), Anchored | Nucleic acid | Sigma–Aldrich | O4387 | Included during communication with authors. Original was Oligo dT (24). Original brand not specified |
| dNTPs (10 mM) | Chemical | Sigma–Aldrich | D7295 | Included during communication with authors. Original brand not specified |
| 25 mM MgCl$_2$ | Chemical | Sigma–Aldrich | M8787 (part of Sigma–Aldrich D4545 from Protocol 1) | Included during communication with authors. Original brand not specified |
| Superscript II (with tube of 5× buffer and 100 mM DTT) | Enzyme | Life Technologies | 18064-014 | Included during communication with authors |
| RNase inhibitor | Enzyme | Sigma–Aldrich | R1274 | Included during communication with authors. Original was RNasin |
| POSTN 5′ primer | Nucleic acid | Sequences provided by original authors; specific brand information will be left up to the discretion of the replicating lab and recorded later | | |
| POSTN 3′ primer | Nucleic acid | | | |
| GAPDH 5′ primer | Nucleic acid | | | |
| GAPDH 3′ primer | Nucleic acid | | | |
| Power SYBR green PCR master mix | Buffer | Life Technologies | 4368577 | – |
| Real-time PCR system | Equipment | Applied Biosystems StepOne | – | Original was from Roche or a StepOnePlus from Applied Biosystems |

## Procedure

Note:

- These metastatic positive lungs from *MMTV-PyMT*[+/tg]; *Postn*[+/+] female mice come from Protocol 1.

1. Mince lungs with bended scissors to smooth paste without any clumps and transfer to tube.
   a. Keep each set of lungs separate (do not pool).
2. Incubate tissue in 6× volume of digestion solution for 1 hr at 37°C with the tube horizontal and shaking at 100 rpm.
   a. Digestion solution: HBSS supplemented with 0.4 U/ml liberase TM, 0.4 U/ml liberase TH, and 25 µg/ml DNase.
      i. Liberase TM: stock solution = 26 U/ml = 5 mg/ml; store at −20°C; use at 1:66 dilution.
      ii. Liberase TH: stock solution = 26 U/ml = 5 mg/ml; store at −20°C; use at 1:66 dilution.
      iii. DNase: stock solution = 10 mg/ml in PBS; store at −20°C; use at 1:400 dilution.
3. Pellet cells at 180×*g* for 5 min at room temperature.
4. Resuspend cells in cold MACS buffer and filter through 100-µm cell strainer using a rubber tip of 2.5 ml syringe to smash remaining tissue pieces.
   a. MACS buffer: 2 mM EDTA in PBS supplemented with 0.5% BSA.
5. Wash strainer extensively with MACS buffer to collect all cells and pellet cells at 180×*g* for 5 min.
6. Wash twice in MACS buffer, pelleting cells at 180×*g* for 5 min between washes.
7. Pellet cells at 180×*g* for 5 min, wash once in FACS buffer and pellet cells at 180×*g* for 5 min.
   a. FACS buffer: 3% FBS in PBS.
8. Resuspend up to $5 \times 10^7$ cells total in FACS buffer at $2 \times 10^7$ cells/ml in polypropylene (opaque) FACS tubes.
9. Either add antibodies directly, or add antibody dilution mixes, and incubate on ice for 30 min in the dark (if staining high amount of cells put on roller at 4°C).
   a. CD45-PE·Cy5.5 (use at 1:300 dilution).
   b. CD31-Pac.Blue (use at 1:200 dilution).
   c. CD34-PE (use at 1:50 dilution).

 d. Include an unstained control for gating.

 e. Include isotype control antibody stains.

 i. Rat IgG2b, κ—PE-Cy5.5.

 ii. Rat IgG2a, κ—Pac.Blue.

 iii. Rat IgG2a, κ—PE.

10. Pellet cells at 180×$g$ for 5 min, wash once with 4 ml FACS buffer, and pellet cells at 180×$g$ for 5 min (if staining high amount of cells perform another wash).

11. Resuspend cells in 500 µl FACS buffer and filter through filter-membrane into polypropylene (opaque) FACS tubes protected from light.

12. Just before FACS add viability dye.

 a. 7AAD (use at 1:1000 dilution).

13. Perform FACS analysis on cells.

 a. Gate for viability (7-AAD$^-$), then gate and collect the different populations to be analyzed:

 i. CD34$^+$/CD31$^-$ cells.

 ii. CD31$^+$ cells.

 iii. CD45$^+$ cells.

 iv. Use negative controls (unstained and isotype control antibodies) to determine gating of populations.

14. Isolate RNA from each collected cell population using TRI reagent following manufacturer's instructions.

 a. Quantify RNA concentrations in each sample using a spectrometer.

 i. Record sample purity ($A_{260/280}$ and $A_{260/230}$ ratios).

15. Reverse transcribe RNA:

 a. cDNA synthesis:

| | |
|---|---|
| Total RNA | 1 ng–5 µg |
| oligodT(18) | 40 pmol |
| oligodT(24) | 40 pmol |
| 10 mM dNTPs | 1.0 µl |
| H$_2$O | Bring up to 12.5 µl |

 b. Heat to 70°C for 5 min, chill on ice for 2 min, then add:

| | |
|---|---|
| 5× superscript II buffer | 4.0 µl |
| 100 mM DTT | 2.0 µl |
| RNasin | 0.5 µl |

 c. Incubate at 45°C for 2 min, then add:

| | |
|---|---|
| Superscript II | 1.0 µl |

 d. Incubate for 1 hr at 42°C.

 e. Heat-inactivate at 70°C for 15 min.

16. Prepare samples in technical duplicates with two dilutions of cDNA (1:25 and 1:125) using POSTN and GAPDH primers and the Power SYBR green PCR Master Mix. Use GAPDH as control.

 a. Primers:

 i. POSTN 5′ primer: AAT GCT GCC CTG GCT ATA TG.

 ii. POSTN 3′ primer: GTA TGA CCC TTT TCC TTC AA.

 iii. GAPDH 5′ primer: CAA GCT CAT TTC CTG GTA TGA CAA T.

 iv. GAPDH 3′ primer: GTT GGG ATA GGG CCT CTC TTG.

 b. Set up SYBR mix (contains 1 mM MgCl$_2$): 10 µl of 1a into 1b (store on ice in the dark).

c. Set up PCR master mix (per reaction):

| | |
|---|---|
| Forward primer | 1.0 µl of 5 µM (5 pmol/µl) |
| Reverse primer | 1.0 µl of 5 µM (5 pmol/µl) |
| $MgCl_2$ | 0.4 µl (for 2 mM) |
| $H_2O$ | Bring up to 4.0 µl |
| SYBR mix | 1.0 µl |

 d. Add 5.0 µl of diluted cDNA and 5.0 µl of PCR master mix to Light cycler capillaries, spin down, and run quantitative PCR reaction following manufacturer's instructions.
 i. Include negative control (no cDNA).
 e. Analyze and compute $\Delta\Delta C_T$ values.

## Deliverables

- Data to be collected:
  - All FACS plots in gating scheme (including all controls), leading to final population of viable, $CD34^+/CD31^-$, $CD31^+$, and $CD45^+$ cells.
  - Purity ($A_{260/280}$ and $A_{260/230}$ ratios) and concentration of isolated total RNA from cells.
  - Raw $C_T$ qRT-PCR values and $\Delta\Delta C_T$ (the $C_T$ value of *POSTN* normalized to *GAPDH*).
  - Graph of *POSTN* normalized expression ($\Delta\Delta C_T$) for each condition. (Compare to Figure 2H).

## Confirmatory analysis plan

This replication attempt will perform the following statistical analysis listed below:

- Statistical analysis:

 Note: At the time of analysis, we will perform the Shapiro–Wilk test and generate a quantile–quantile (q–q) plot to assess the normality of the data and also perform Levene's test to assess homoscedasticity. If the data appear skewed, we will perform the appropriate transformation in order to proceed with the proposed statistical analysis. If this is not possible, we will perform the equivalent non-parametric test.

  - One-way ANOVA of *POSTN* RNA expression in $CD34^+/CD31^-$, $CD31^+$, and $CD45^+$ cells.
    - Planned comparisons with the Bonferroni correction:
      - $CD34^+/CD31^-$ vs $CD45^+$.
      - $CD31^+$ vs $CD45^+$.
- Meta-analysis of effect sizes:
  - Compute the effect sizes of each comparison, compare them against the reported effect size in the original paper and use a meta-analytic approach to combine the original and replication effects, which will be presented as a forest plot.

## Known differences from the original study

All known differences are listed in the materials and reagents section above with the originally used item listed in the comments section. All differences have the same capabilities as the original and are not expected to alter the experimental design.

## Provisions for quality control

Negative staining and isotype controls are included to assess antibody staining relative to background during FACS analysis. The sample purity ($A_{260/280}$ and $A_{260/230}$ ratios) of the isolated RNA from each sample will be reported. All data obtained from the experiment—raw data, data analysis, control data, and quality control data—will be made publicly available, either in the published manuscript or as an open access data set available on the Open Science Framework project page for this study (https://osf.io/vseix).

## Power calculations

### Protocol 1
Summary of original data (estimated from Figure S13).

| Figure 3B and S13: Number of metastases or size of primary tumor | Mean | SD | N |
|---|---|---|---|
| Number of metastases in MMTV-PyMT; Postn$^{+/+}$ mice | 15.78 | 17.54 | 18 |
| Number of metastases in MMTV-PyMT:Postn$^{-/-}$ mice | 2.765 | 5.069 | 17 |
| Size of primary tumor in MMTV-PyMT; Postn$^{+/+}$ mice | 1.221 | 0.6023 | 18 |
| Size of primary tumor in MMTV-PyMT; Postn$^{-/-}$ mice | 1.186 | 0.5901 | 16 |

### Size of primary tumor
Test family

- 2-tailed *t*-test, difference between two independent means, alpha error = 0.05.

  Sensitivity calculations performed with G*Power software, version 3.1.7 (*Faul et al., 2007*).

| Group 1 | Group 2 | Effect size *d* | A priori power | Group 1 sample size | Group 2 sample size |
|---|---|---|---|---|---|
| MMTV-PyMT; Postn$^{+/+}$ mice | MMTV-PyMT; Postn$^{-/-}$ mice | 1.145371* | 80.0% | 13 | 13 |

*This excludes one outlier data point (2.83) from the *Postn*$^{-/-}$ data.

### Number of metastases
Test family

- 2-tailed *t*-test, difference between two independent means, alpha error = 0.05.

  Power Calculations performed with G*Power software, version 3.1.7 (*Faul et al., 2007*).

| Group 1 | Group 2 | Effect size *d* | A priori power | Group 1 sample size | Group 2 sample size |
|---|---|---|---|---|---|
| MMTV-PyMT; Postn$^{+/+}$ mice | MMTV-PyMT; Postn$^{-/-}$ mice | 1.186517* | 82.7% | 13 | 13 |

*This excludes one outlier data point (61) from the *Postn*$^{+/+}$ data and two outlier data points (18 and 13) from the *Postn*$^{-/-}$ data.

Test family

- Negative binomial regression, alpha error = 0.05.

  Analysis of original data: performed with R software, version 3.1.2 (*R Core Development Team, 2014*).
  Chi-square goodness of fit test, p-value = 0.2535.
  Regression coefficient, Genotype (*Postn*$^{-/-}$) = −1.742, incident rate ratio = 0.1752, p-value = 0.000923.
  Predicted values from model:

| Data set being analyzed | Mean | SE |
|---|---|---|
| Number of metastases in MMTV-PyMT; Postn$^{+/+}$ mice | 15.78 | 5.594 |
| Number of metastases in MMTV-PyMT:Postn$^{-/-}$ mice | 2.765 | 1.073 |

Power Calculations performed with R software, version 3.1.2 (*R Core Development Team, 2014*).

| Groups | Number of simulations | A priori power | Sample size |
|---|---|---|---|
| Number of metastases in MMTV-PyMT; *Postn*$^{+/+}$ mice and MMTV-PyMT; *Postn*$^{-/-}$ mice | 10,000* | 81.2% | 12 per group |

*The original data were randomly sampled from, with replacement, to create simulated data sets. For a given *n* (the number of observations) 10,000 simulations were run and the Chi-square goodness of fit test and regression coefficient (Genotype (*Postn*$^{-/-}$)) was calculated for each simulated data set. Any model fit with p < 0.05 was excluded. The power was then calculated by counting the number of times p ≤ 0.05 and dividing by the number of model fits.

## Protocol 2

Summary of original data (estimated from Figure 2H).

| Figure 2H: qPCR analysis of POSTN expression | N | Mean | SD |
|---|---|---|---|
| CD34$^{+}$/CD31$^{-}$ pulmonary fibroblasts | 3 | 1.7 | 0.6 |
| CD31$^{+}$ endothelial cells | 3 | 0.15 | 0.1 |
| CD45$^{+}$ immune cells | 3 | 0.01 | 0.04 |

Test family

■ ANOVA: Fixed effects, omnibus, one-way, alpha error = 0.05.

Power calculations performed with G*Power software, version 3.1.7 (*Faul et al., 2007*).
ANOVA F test statistic and partial η$^2$ performed with R software, version 3.1.2 (*R Core Development Team, 2014*).

| Groups | F test statistic | Partial η$^2$ | Effect size *f* | A priori power | Total sample size |
|---|---|---|---|---|---|
| CD34$^{+}$/CD31$^{-}$, CD31$^{+}$, and CD45$^{+}$ | F(2,6) = 21.306 | 0.876574 | 2.664962 | 89.8%* | 6* (3 groups) |

*A total sample size of 18 will be used based on the planned comparison calculations making the power 99.9%.

Test family

■ 2-tailed *t*-test, difference between two independent means, Fisher's LSD test: alpha error = 0.05.

Power calculations performed with G*Power software, version 3.1.7 (*Faul et al., 2007*).

| Group 1 | Group 2 | Effect size d | A priori power | Group 1 sample size | Group 2 sample size |
|---|---|---|---|---|---|
| CD34$^{+}$/CD31$^{-}$ | CD45$^{+}$ | 3.974546 | 94.6%* | 3* | 3* |
| CD31$^{+}$ | CD45$^{+}$ | 1.838290 | 81.8% | 6 | 6 |

*6 tumors will be used per group based on the CD31$^{+}$ to CD45$^{+}$ comparison making the power 99.9%.

## Acknowledgements

The Reproducibility Project: Cancer Biology core team would like to thank the original authors, in particular Joerg Huelsken and Ilaria Malanchi, for generously sharing critical information as well as reagents to ensure the fidelity and quality of this replication attempt. We thank Courtney Soderberg at the Center for Open Science for assistance with statistical analyses. We would also like to thank the following companies for generously donating reagents to the Reproducibility Project: Cancer Biology;

American Type Culture Collection (ATCC), BioLegend, Charles River Laboratories, Corning Incorporated, DDC Medical, EMD Millipore, Harlan Laboratories, LI-COR Biosciences, Mirus Bio, Novus Biologicals, Sigma–Aldrich, and System Biosciences (SBI).

## Additional information

### Group author details

**Reproducibility Project: Cancer Biology**

Elizabeth Iorns: Science Exchange, Palo Alto, California; William Gunn: Mendeley, London, United Kingdom; Fraser Tan: Science Exchange, Palo Alto, California; Joelle Lomax: Science Exchange, Palo Alto, California; Nicole Perfito: Science Exchange, Palo Alto, California; Timothy Errington: Center for Open Science, Charlottesville, Virginia

### Competing interests

FI: This is a Science Exchange associated lab. MMD: This is a Science Exchange associated lab. NI: This is a Science Exchange associated lab. AC: This is a Science Exchange associated lab. RP:CB: EI, FT, JL, and NP are employed by and hold shares in Science Exchange Inc. The other authors declare that no competing interests exist.

### Funding

| Funder | Author |
| --- | --- |
| Laura and John Arnold Foundation | Reproducibility Project: Cancer Biology |

The Reproducibility Project: Cancer Biology is funded by the Laura and John Arnold Foundation, provided to the Center for Open Science in collaboration with Science Exchange. The funder had no role in study design or the decision to submit the work for publication.

### Author contributions

FI, MMD, NI, AC, EG, MAL, Drafting or revising the article; RP:CB, Conception and design, Drafting or revising the article

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
