## [Decision Letter]

Thank you for sending your work entitled “Registered report: Interactions between cancer stem cells and their niche govern metastatic colonization” for consideration at *eLife*. Your article has been evaluated by Janet Rossant (Senior editor), Gordana Vunjak-Novakovic°(Reviewing editor), and three reviewers, one of whom, M Dawn Teare, has agreed to share her identity.

The Reviewing editor and the reviewers discussed their comments before we reached this decision, and the Reviewing editor has assembled the following comments to help you prepare a revised submission.°

This Registered report is part of a bigger project named Reproducibility Project: Cancer Biology, which aims to address concerns about reproducibility of scientific data published in the field of cancer research between 2010 and 2012. Within this project, parts of the experiments from 50 selected publications will be reproduced by an independent laboratory.°

The report describes the plan for the reproduction of two selected experiments from the original article “Interactions between cancer stem cells and their niche govern metastatic colonization” by Malanchi et al. in Nature in 2012 (10). The first experiment is to measure metastatic lung cancer in MMTV-PyMT transgenic mice crossed with the POSTN Knock out mice, and controls, with focus on the role of periostin in metastatic progression.°

The second experiment is to isolate various cell types from lungs harboring metastasis to confirm the cellular source of periostin in response to metastatic growth. We find the data of high quality and possibly of interest to the scientific community. However, the reviewers also made some comments that we would like you to address.°

1) The key finding that periostin acts as a fundamental stromal factor for metastatic progression has been meanwhile experimentally validated in several independent labs by using other models. In light of this, it would be helpful to explain in more detail what exactly is your reproducibility project providing in terms of scientific and technical validation.°

2) The two repeated experiments analyze the source of POSTN expression in the lung and whether it affects the number/size of primary and secondary tumor formation in a spontaneous mouse model of breast cancer (MMTV-PyMT). Please provide justification for these particular choices of experiments. The reviewers suggest starting the manuscript by an overview of the original results and experimental designs, and the reasoning for repeating the two selected experiments. This would help the reader, who may not be familiar with the original work, to better evaluate the scope and focus of this repetition.°

3) Discussing the results of the original study, reproduced here, in the context of other work published over the last few years, would also be very helpful.°

4) Finally, the use of statistics requires some explanation. For protocol 1, the authors should discuss their treatment of outliers: why such outliers have arisen and how they can be best handled in the analysis. For protocol 2, very small numbers in each group are planned and this is due to assuming the data is normally distributed. The normality of data should be addressed in the analysis and if data is not clearly normally distributed it would be fair to relax the normality assumption and check that conclusions are not changed if a non-parametric comparison is used.°

---

## [Author Response]

*1) The key finding that periostin acts as a fundamental stromal factor for metastatic progression has been meanwhile experimentally validated in several independent labs by using other models. In light of this, it would be helpful to explain in more detail what exactly is your reproducibility project providing in terms of scientific and technical validation*.*°*

This project aims to evaluate the predictors of directly reproducing a subset of the published literature. Thus, the focus is on a collection of experimental outcomes, and the factors associated with them, and not the conclusions from any given paper, which are based on multiple experiments and models.

Additionally, the project is focused on direct replications (same methodology/system) compared to conceptual replications (similar experiment, but different techniques/models). Conceptual replication is as vital for gaining understanding of an effect as direct replication is for increasing confidence that the effect occurs. The focus on understanding if the effects drawn from a single model can be reproduced will provide a means to understand the challenges and predictors of reproducing any given experiment based on current research and reporting practices. While this does not speak to the robustness of the effect, such as can be inferred through multiple models/approaches, it does provide a mechanism to examine the extent to which an effect with a given model can be observed again. We will also limit the conclusions that can be drawn to only this model.

*2) The two repeated experiments analyze the source of POSTN expression in the lung and whether it affects the number/size of primary and secondary tumor formation in a spontaneous mouse model of breast cancer (MMTV-PyMT). Please provide justification for these particular choices of experiments. The reviewers suggest starting the manuscript by an overview of the original results and experimental designs, and the reasoning for repeating the two selected experiments. This would help the reader, who may not be familiar with the original work, to better evaluate the scope and focus of this repetition*.*°*

Thank you for the suggestion. We have included an additional paragraph to describe the original work and the rational for the two selected experiments included.

*3) Discussing the results of the original study, reproduced here, in the context of other work published over the last few years, would also be very helpful*.*°*

We have expanded the Introduction to include other published studies in the context of the original study.

*4) Finally, the use of statistics requires some explanation. For protocol 1, the authors should discuss their treatment of outliers: why such outliers have arisen and how they can be best handled in the analysis. For protocol 2, very small numbers in each group are planned and this is due to assuming the data is normally distributed. The normality of data should be addressed in the analysis and if data is not clearly normally distributed it would be fair to relax the normality assumption and check that conclusions are not changed if a non-parametric comparison is used*.*°*

For protocol 1, we’ve included language to discuss the detection of outliers and plan to perform the test with and without the outliers to ascertain if a difference occurs in the analysis. The original analysis appears to have excluded the outliers in the statistical test they used. Additionally, we have further explored the original estimated data and the metastatic foci counts violate the normality and homoscedasticity assumption of the test used, as determined by the Shapiro–Wilk test (and Q–Q plots) and Levene’s test. So, in addition to a Student’s *t*-test (which was originally performed) we plan to analyze the data using negative binomial regression. The power calculations were also performed again to ensure the sample size was still adequate for this test.

For protocol 2, we have added language to the manuscript to clarify that we will perform tests for normality and homoscedasticity.